# Tuning Easy Magnetization Direction and Magnetostatic Interactions in High Aspect Ratio Nanowires

**DOI:** 10.3390/nano11113042

**Published:** 2021-11-12

**Authors:** Hafsa Khurshid, Rahana Yoosuf, Bashar Afif Issa, Atta G. Attaelmanan, George Hadjipanayis

**Affiliations:** 1Department of Applied Physics and Astronomy, University of Sharjah, Sharjah 27272, United Arab Emirates; ryoosuf@sharjah.ac.ae (R.Y.); aattaelmanan@sharjah.ac.ae (A.G.A.); 2Department of Radiology, Dartmouth Hitchcock Medical Center, Lebanon, NH 03766, USA; 3Department of Medical Diagnostic Imaging, University of Sharjah, Sharjah 27272, United Arab Emirates; bissa@sharjah.ac.ae; 4Department of Physics and Astronomy, University of Delaware, Newark, DE 19716, USA; hadji@udel.edu

**Keywords:** cobalt nanowires, magnetic interactions, electrochemical deposition

## Abstract

Cobalt nanowires have been synthesized by electrochemical deposition using track-etched anodized aluminum oxide (AAO) templates. Nanowires with varying spacing-to-diameter ratios were prepared, and their magnetic properties were investigated. It is found that the nanowires’ easy magnetization direction switches from parallel to perpendicular to the nanowire growth direction when the nanowire’s spacing-to-diameter ratio is reduced below 0.7, or when the nanowires’ packing density is increased above 5%. Upon further reduction in the spacing-to-diameter ratio, nanowires’ magnetic properties exhibit an isotropic behavior. Apart from shape anisotropy, strong dipolar interactions among nanowires facilitate additional uniaxial anisotropy, favoring an easy magnetization direction perpendicular to their growth direction. The magnetic interactions among the nanowires were studied using the standard method of remanence curves. The demagnetization curves and Delta m (Δm) plots showed that the nanowires interact via dipolar interactions that act as an additional uniaxial anisotropy favoring an easy magnetization direction perpendicular to the nanowire growth direction. The broadening of the dipolar component of Δm plots indicate an increase in the switching field distribution with the increase in the nanowires’ diameter. Our findings provide an important insight into the magnetic behavior of cobalt nanowires, meaning that it is crucial to design them according to the specific requirements for the application purposes.

## 1. Introduction

Over the last two decades, much attention has been paid to the fabrication of one-dimensional (1D) nanostructures, including nanotubes, nanowires, nanofibers, and nanobelts [1]. Owing to their shape anisotropy, these materials show interesting properties compared to their bulk and spherical counterparts and demonstrate enormous potential in the field of magnetic, electronic, and optoelectronic devices, catalysis, sensors, and spintronic devices [2,3]. In 1D nanostructure arrays such as nanorods, nanowires and nanotubes, apart from the shape anisotropy, a strong dipolar coupling between them also contributes to their magnetic behavior which can be modulated by the nanowires’ packing density [4]. This coupling acts as an additional uniaxial anisotropy favoring the easy magnetization direction (written henceforth as easy direction) perpendicular to the nanowire growth direction, as evident by the ferromagnetic resonance spectra and hysteresis loops [1,5,6].

In an array of magnetic nanowires, the overall contribution to energy depends on (1) their magnetocrystalline anisotropy, (2) dipolar interactions, and (3) shape anisotropy. The magnetocrystalline anisotropy is an intrinsic property of a material, whereas shape anisotropy depends upon the nanowire aspect ratio and plays an important role to the overall magnetic behavior of nanowires [7]. Shape anisotropy favors easy direction along the nanowire growth direction, whereas the magnetic packing density (magnetic volume fraction or magnetic packing fraction) plays a major role in tuning their dipolar interactions [8]. Assuming that all pores are uniformly filled up by the nanowires, the magnetic packing density is quantitatively equal to the membrane porosity. Therefore, interaction effects are of key importance for tuning the magnetic properties since they contribute to the total energy of the system and may drastically affect the switching field distribution, super ferromagnetic collective behavior, as well as the heating efficiency during magnetic fluid hyperthermia [9,10]. It is imperative to tailor the nanowire physical dimensions to match the specific requirements for each kind of applications. Much effort has been focused in tuning the nanowire spacing-to-diameter ratio, and hence in tailoring their magnetic properties accordingly. However, from the applications point of view it is equally crucial to estimate the magnetic interactions and switching field distribution in high-density arrays quantitatively. Magnetic interactions directly impact the switching field distribution [11,12]. For the applications point of view, it is important to estimate the interaction field among the nanowires. Such interaction can be explored by analyzing nanowire remanence curves, such as DC demagnetization (DCD) and isothermal remanence (IRM) curves. Corradi and Wohlfarth [13] introduced the interaction field factor as the difference between the fields where the DCD curve is zero (H_0d_) and where the IRM curve is 1/2 (H_0.5r_) normalized by H_0d_. In the absence of interactions, the two curves will be equal, but with interactions they are different, and the difference can be measured. However, this difference provides only a qualitative information about the interaction field [14]. Araujo et al. have quantitatively estimated the interaction field among arrays of nanowires of different packing density [10]. The field difference between the IRM and DCD remanence curves at a normalized magnetization value (M/M_s_) of m = 1/3 is proportional to the interaction field. M_s_ is nanowires magnetization at maximum field applied. For non-interacting nanowires assembly, both remanence curves intersect at m = 1/3. However, if the nanowires are interacting, the curves intersect at a different magnetization value resulting in a measurable field difference at m = 1/3. With a decrease in the nanowire’s diameter, the packing density increases, and hence the interaction field increases.

In this work, we have synthesized hcp cobalt nanowires in AAO templates with varying diameters and spacing. Room temperature magnetic properties showed that the nanowire easy direction can be tuned parallel or perpendicular to the nanowire growth direction by varying the nanowire spacing-to-diameter ratio. The nanowire interactions were probed from the demagnetization and remanence curves using the standard delta M plots. Moreover, interaction field among nanowires was estimated using a model developed by Araujo et al. [10]. The interaction field increased with nanowire magnetic packing density that is equal to template porosity, assuming all pores are uniformly filled. However, after a certain value of packing density, the interaction field does not increase any further, rather this trend is reversed. At this packing density, the nanowire spacing-to-diameter ratio is small enough that nanowires exhibit isotropic magnetic behavior, as evidenced from their hysteresis loops. When the nanowire diameter is increased, the broadening of the dipolar component of Δm plots indicate an increase in the switching field distribution. Our findings provide insight into the variation of nanowire interaction field along with nanowire spacing and diameter. It is anticipated that beside the interwire spacing and diameter, the interaction field strongly depends on the volume of individual nanowires. Our findings provide important insight into the magnetic behavior of cobalt nanowires that is crucial to design them according to the specific requirements for the application purpose.

## 2. Materials and Methods

### 2.1. Materials Synthesis

The nanowires were grown in situ by electrodepositing cobalt over the AAO templates. These templates got arrays of empty columns whose diameter and spacing can be varied during template fabrication. They provide opportunity to tune the nanowire diameter, spacing and center to center spacing [15]. Electrodeposition facilitates nanomaterial fabrication at low cost and energy over other conventionally used methods such as molecular beam epitaxy and lithography [16]. Control over the electrodeposition parameters (current density, bath acidity, etc.), gives the ability to tune the magnetic and microstructure properties of nanowires [17].

The AAO template were prepared by using a well-established method, the two-step anodization procedure [18]. Before anodizing, high purity (99.999%) aluminum foil was degreased in acetone and etched by sodium hydroxide (NaOH) to remove aluminum oxide surface layer [1,6,16,17]. The aluminum foil was anodized under constant voltage of 40 V in 0.3 M C_2_H_2_O_4_ at 0 °C. The remaining aluminum was removed using a copper chloride solution. A subsequent etching was carried out in a phosphoric acid solution at 40 °C to remove the barrier on the bottom side of the AAO template and to widen the pores slightly. This process resulted in AAO template with hollow columns (pores) along the template thickness. The pore density and pore diameter strongly depend upon the anodization voltage, anodizing agent pH (acid concentration) and temperature [15]. The pore diameter was adjusted by varying the anodization voltage from 25 V to 40 V, as reported earlier in the literature [15]. To obtain pores of diameter smaller than 30 nm, anodization was performed at 20 V using 0.5 M sulfuric acid. Table 1 lists anodization conditions and pore diameter for AAO template fabrication. The anodization was carried out overnight at 0 °C. The time and temperature were kept constant for all the samples. For the very large pore diameter (188 nm, sample H4), a commercially available Whatman AAO template was used. The porosity of the template depends upon the interpore distance (average center to center spacing between pores). Higher interpore distance translates to a smaller density of pores per surface unit, which in turn depends upon electrolyte concentration. Table 2 lists the samples used in this study.

A conventional electrochemical cell was used to fabricate Co nanowires. Before deposition, a 100 nm thick layer of copper was sputtered at one side to be used as the working electrode. A graphite rod was used as the counter electrode. The electrolyte was composed of 1.25 moles of cobalt sulphate CoSO_4_ and 0.65 moles of boric acid H_3_BO_3_. The electrolyte pH was adjusted to 4, using diluted H_2_SO_4_. The boric acid helps to enhance current efficiency and suppress metal-hydroxide formation [19]. Electrodeposition was performed at room temperature at a constant current density of 25 A/m^2^ room temperature. The current density was estimated by accounting the total area of the membrane surface exposed to the electrolyte. The nanowires’ length was kept equal by using the same deposition rate and controlling deposition time. A schematic illustration of the aluminum anodization and cobalt nanowire deposition is shown in Figure 1A.

### 2.2. Analytical Methods

The morphology and microstructure of the AAO templates and nanowires were observed by scanning electron microscopy (SEM), transmission electron microscopy ((TEM, JEOL JEM-3010), and by X-ray diffraction (XRD, Rigaku Ultima IV) with CuKα radiation. The magnetic properties were measured by a vibrating sample magnetometer (VSM) and Superconducting Quantum Interferometer Device (SQUID) with the applied field either perpendicular or parallel to the nanowire direction of growth.

## 3. Results and Discussion

Figure 2 shows a cross sectional SEM image and top view of electrodeposited nanowires within the templates. The lines of lighter contrast intruding the anodized alumina from the top surface in uniform columns are the nanowires. Across the entire sample, the nanowire length is the same, as evidenced by the sharp interface between the nanowire-filled pores and the unfilled region below. A top view of the AAO template indicates its honeycomb-like structure with pores in a hexagonal arrangement. HRTEM analysis reveals the crystalline structure of nanowires. The lattice fringes exhibit a spacing of 2.03 Å, which corresponds to (200) planes of hcp Co. It is to be reminded that TEM analysis does not reflect preferential crystallographic growth, as it was performed on the nanowires after removing the AAO template.

The XRD analysis of Co nanowire arrays (within AAO template) provides further insight into their crystalline structure. The XRD micrographs (Figure 3) match with the hexagonal phase of the standard Co powder pattern (PDF#05-0727), indicating that nanowires possess hcp structure. It has been reported that during synthesis, if the pH value of electrolyte is above 3, the Co nanowires exhibit hcp crystallographic structure [19]. A strong hcp-(100) peak at 41.6° indicates that Co nanowires have a preferred orientation along (100) the direction that is perpendicular to nanowire growth direction. Besides the (100) peak, in each sample the other peaks are very small, if present at all. For all the samples, deposition current density (accounting AAO area exposed to the electrolyte) was kept constant. It is to be noted, that all the samples showed preferred growth direction along (100), irrespective of nanowire diameter and spacing. A broad peak located in the 2θ angle range of (20–30°) depicting amorphous nature of AAO template. It is to be noted that the sharp diffracted peaks points to the fact nanowires are composed of bigger grains (above 100 nm); therefore, it is not possible to estimate grain size using Scherrer’s formula.

The room temperature magnetic properties were measured by applying the magnetic field parallel and perpendicular to the nanowire growth direction (out of plan and in plan with AAO templates, respectively). The normalized hysteresis loops for nanowires with spacing-to-diameter ratio (s/d) of 0.74 (sample H1) are shown in Figure 4. These hysteresis loops reveal that the nanowire array exhibits uniaxial magnetic anisotropy with the easy direction parallel to the nanowire growth direction. Figure 4b,c illustrates the normalized hysteresis loops for the samples with spacing to diameter s/d = 0.53 and s/d = 0.37, respectively. Bantu et al. [2] has developed a magnetostatic model based on the competition between the dipolar interactions, demagnetizing field, and magnetocrystalline anisotropy. A crossover of the easy direction is expected with the change in nanowire volume [2]. For the Co hcp at room temperature, this crossover of magnetization would occur when effective magnetic fields parallel ($H_{\mathrm{eff}}∥)$ and perpendicular ($H_{\mathrm{eff}}⊥)$ to the direction of growth magnetic nanowires are same, that is, $H_{\mathrm{eff}}\|={\;H}_{\mathrm{eff}}\;⊥$, corresponding to the critical ratio between the volume V of wires and the distance d so that (V/d^3^) ~ 0.21. When the V/d^3^ < 0.21, the easy direction is parallel to the growth directional of the wires and for V/d^3^ > 0.21 perpendicular to the nanowire growth direction. It was reported that easy direction switches from parallel to perpendicular when the nanowires’ length is increased above 6 µm [2]. Moreover, magnetic-force-microscopy studies on hcp Co nanowires with diameter 90 nm and length over 10 µm, revealed magnetization frustration due to the competition between the magneto-crystalline polarization along the easy direction and the shape anisotropy along the nanowire growth direction^21^. Since all of our samples have a high aspect ratio (above 300) with V/d^3^ > 0.21, the magnetization easy direction is expected to be perpendicular to nanowires. However, the easy direction varies with the nanowire spacing to diameter-to-spacing ratio, as seen in Figure 4. Magnetic behavior of the nanowires array is governed by the balance between different energy contributions from (1) shape anisotropy of individual nanowires, (2) magnetocrystalline anisotropy induced by the texture, and (3) the magnetostatic dipolar interactions among the nanowires [20,21]. For the nanowires with a larger spacing-to-diameter ratio (s/d = 0.74) the magnetic data reveal an easy direction along the nanowire growth direction which is parallel to the applied field. It is believed that the preferred easy direction along the nanowire growth direction is due to shape anisotropy along that direction. However, below a certain s/d value, the dipolar interactions are strong enough to overcome shape anisotropy, thus favoring easy direction perpendicular to the nanowire growth direction. Intriguingly, the easy direction switches from the parallel to the perpendicular direction, as seen in Figure 4b with s/d ratio of 0.53. Peculiarly, the nanowires magnetization does not show anisotropic behavior if the spacing to diameter is reduced further. The magnetization curves are essentially identical, with the applied field perpendicular or parallel to the nanowire growth direction when s/d = 0.39 (Figure 4c). In a closed pack systems, the nanowire magnetic behavior strongly depends upon their packing density^1^, which also equals template porosity given by P = 3.67 (d/s)^2^. Assuming all pores are uniformly filled up by the nanowires, this magnetic packing density is quantitatively equal to the membrane porosity. For sample H1 (s/d = 0.74), the estimated packing density is 5.3%, which increases to 13% and 23% for samples H4 (s/d = 0.53) and sample H5 (s/d = 0.37), respectively. Ideally, P approaches 1 for a continuous film. Varying the spacing-to-diameter ratio and packing density also have an impact on the interactions among the nanowire arrays. For a system of closely packed magnetic nanostructures, magnetostatic coupling plays an important role during the switching process. However, the strength of coupling depends strongly on the physical dimensions such as the length, diameter and spacing [22]. Magnetocrystalline anisotropy and dipolar interactions favor an easy direction perpendicular to the growth direction. However, below a critical s/d ratio in the nanowires, the magnetic packing density is high enough that the nanowire array approaches the magnetic behavior of a bulk material in thin film form, e.g., the magnetization may prefer to lie-down in the membrane plane in order to avoid the large stray field or demagnetizing energy due to the free magnetic poles at the membrane surface [23].

To probe the nanowire interactions, dc demagnetization (DCD) and isothermal remanence (IRM) curves were measured at room temperature. The IRM curve (m_r_) was measured by applying and removing an increasingly positive field. When the field is removed after each field increment, the remanent magnetization is measured along with the final field value of the corresponding increment. The DCD curve (m_d_) starts at the highest positive remanence value after applying and removing a large positive magnetic field, and by applying and removing and increasingly negative field, the system is taken to the final state where remanence has its maximum value but in the negative direction. This measured magnetization curve reflects the irreversible changes in the sample since the measurements are always performed at the zero field. Ideally, for a non-interacting nanoparticle assembly, m_r_ and m_d_ follow the Wohlfarth relation [24],
$$\Delta m={\;m}_{d}-\left(1-2m_{r}\right)=0$$

A negative Δm depicts dominant magnetostatic or dipolar interactions among the system particles/nanowires, whereas positive Δm curves point to the dominant magnetostatic interactions among the nanowires. Typical remanence and Δm curves versus the magnetic field H are shown in Figure 5a. Figure 5b shows Δm plots for samples s/d = 0.74, 0.53, 0.39, respectively. Delta m (Δm) plots revealed the nanowires interact with each other via dominant magnetostatic or dipolar interactions, as all the samples exhibit negative Δm values. It is assumed that each nanowire behaves as a single entity during the switching process, hence favoring their antiparallel alignment that favors nanowires to interact via dipolar interactions. When two samples of same s/d ratios but different diameters and spacing were compared, the broadening of the dipolar component of Δm plots was very different, as seen in Figure 5c. This indicates an increase in the switching field distribution with the increase in the nanowire diameter.

To quantitatively estimate the average value of interaction fields among the nanowires in an array, we used the model developed by Araujo et al. [10]. For a non-interacting assembly, Wohlfarth’s relation requires both remanence curves to intersect at m = 1/3; while in the presence of an interaction, the curves intersect at a different magnetization value (Figure 6). The field difference between the IRM and DCD remanence curves (ΔH_1/3)_ with normalized magnetization value of m = 1/3 is proportional to the interaction field α = 3/2 ΔH_1/3_. Qualitative information about the interaction field were obtained from the position (m_o_) and shift along the magnetization direction of the intersection between the remanence curves (m_r_ and m_d_), with respect to the value of one-third (δm = m_o_ − 1/3). The positive δm value corresponds to ferromagnetic interactions, and negative δm to antiferromagnetic interactions. Nanowires with a smaller diameter exhibit a higher interaction field, as seen in Figure 6. This trend is opposite above a certain diameter of 60 nm and reached above 1000 Oe for the 190 nm-diameter nanowires. These data can be explained by accounting the higher magnetic packing density in the nanowires with a smaller diameter. With a smaller diameter and lower magnetic packing density (below 12%), the individual nanowires behave as macrospins and are more likely to switch irreversibly without altering the magnetic state of surrounding nanowires. However, at larger diameters and higher magnetic packing density, the interaction field is much stronger, as is evidenced from Figure 6.

## 4. Conclusions

In summary, we have prepared arrays of hcp Co nanowires with various diameters and interwire spacings. Their microstructure and crystal structure were studied using electron microscopy and X-ray diffraction. The magnetic properties indicate an isotropic or anisotropic behavior depending on the interwire spacing-to-diameter ratio. The easy direction can be tailored from the parallel to perpendicular direction of the nanowire growth direction. From the demagnetization curves and Δm plots, it is shown that nanowires interact via dipolar coupling that acts as an additional uniaxial anisotropy favoring an easy direction perpendicular to the wire growth direction. The magnetization easy direction can be tuned parallel or perpendicular to the nanowire direction by changing their packing density. The interaction field increases with nanowire packing density, and this trend is reversed after a certain value. At this packing density, the interwire spacing-to-diameter ratio is small enough that the nanowires exhibit isotropic magnetic behavior, as evidenced from their hysteresis loops. When the nanowire diameter is increased, the broadening of the dipolar component of ΔM plots indicates an increase in the switching field distribution. Our findings provide insight into the nanowires’ interaction field variation along with the nanowires’ spacing and diameters. It is anticipated that besides the interwire spacing and diameter, the interaction field strongly depends upon the volume of individual nanowires. Such findings are important to design material according to the specific magnetic properties’ requirements for the application purpose.

## Figures and Tables

**Figure 1 nanomaterials-11-03042-f001:**
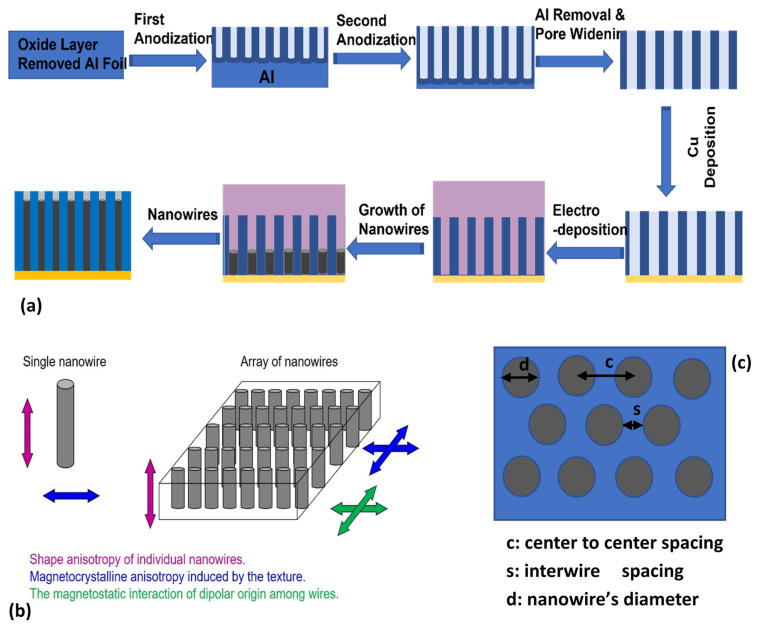
(**a**) A schematic illustration of sample preparation in this study, (**b**) energy contribution to magnetic behavior in nanowires’ array, and (**c**) spacing and diameter symbols used in the text are illustrated in ‘c’.

**Figure 2 nanomaterials-11-03042-f002:**
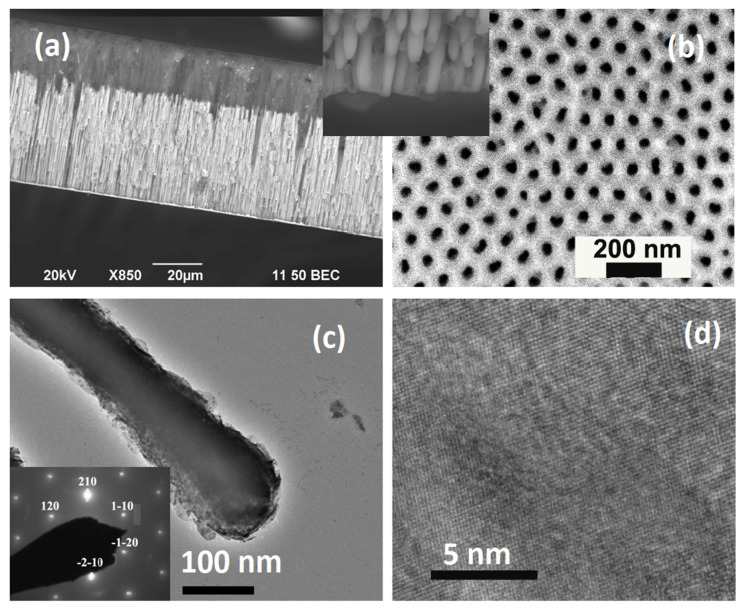
SEM image of AAO template (**a**) cross-sectional and (**b**) top view. Bright field TEM image of one the wires; scalebar 200 nm (**c**) along with selected area diffraction and (**d**) high-resolution TEM with d-spacing (2.03 Å) corresponds to (200) planes of hcp Co.

**Figure 3 nanomaterials-11-03042-f003:**
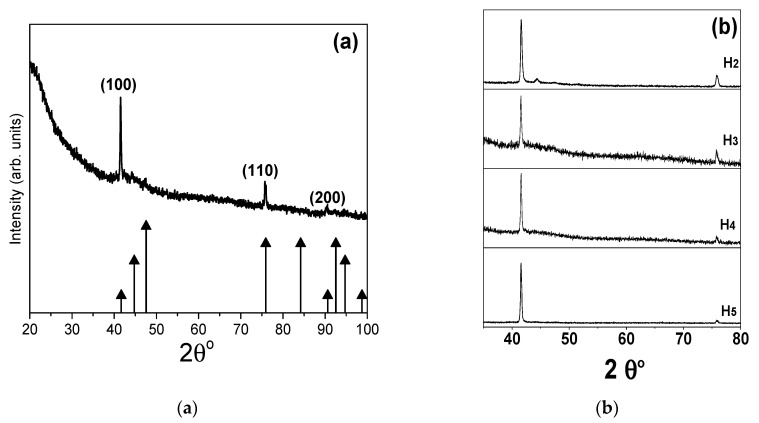
XRD micrographs of sample H1 (**a**) and all other samples (H2, H3, H4, and H5) used in this study (**b**), along with standard hcp cobalt XRD pattern.

**Figure 4 nanomaterials-11-03042-f004:**
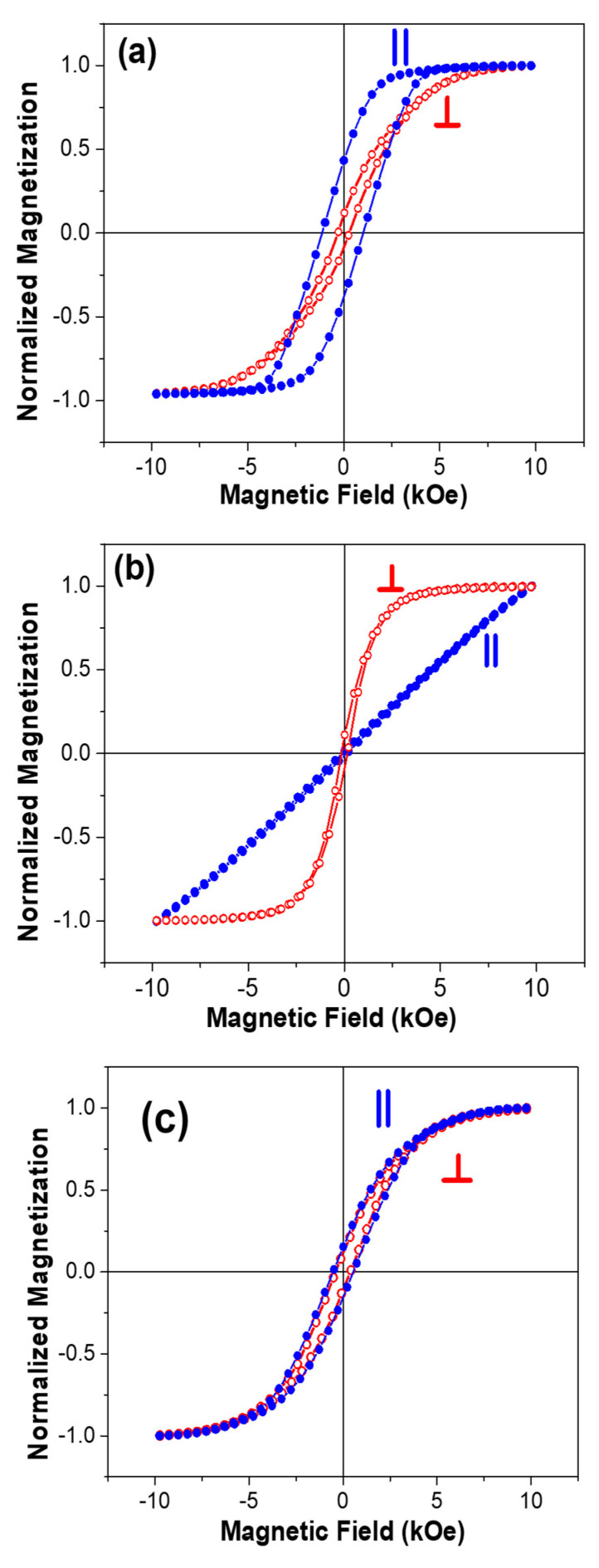
Room temperature normalized magnetization (M/Ms) dependence upon magnetic field with spacing-to-diameter ratio (**a**) 0.74, (**b**) 0.53 and (**c**) 0.37.

**Figure 5 nanomaterials-11-03042-f005:**
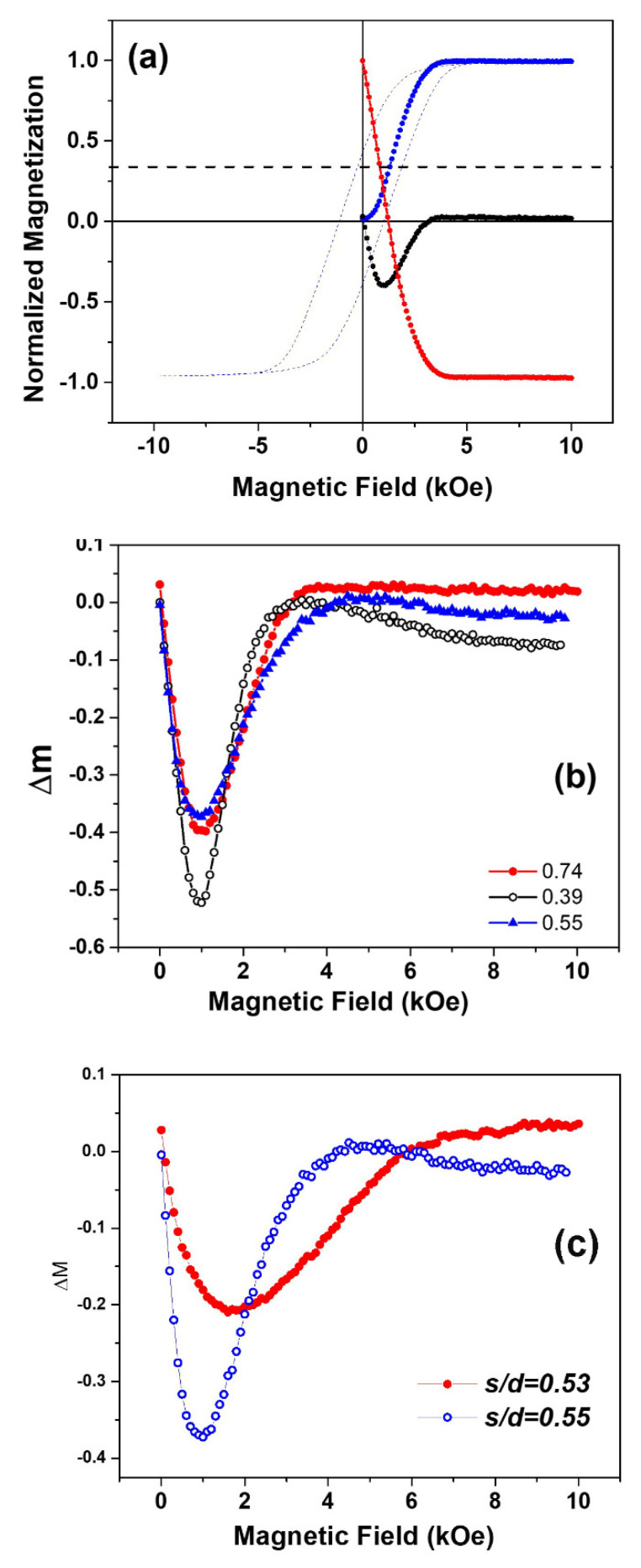
(**a**) IRM and DCD remanence curves, m_r_ (blue) and m_d_ (red), respectively, along with the major hysteresis loop (dotted red) and calculated Δm curve (black) for the nanowires sample with s/d = 0.61 magnetic packing density 10.2%. (**b**) the Δm curves with different spacing-to-diameter ratios, as mentioned. (**c**) Δm curves of the same s/d ratios and packing density but different diameters.

**Figure 6 nanomaterials-11-03042-f006:**
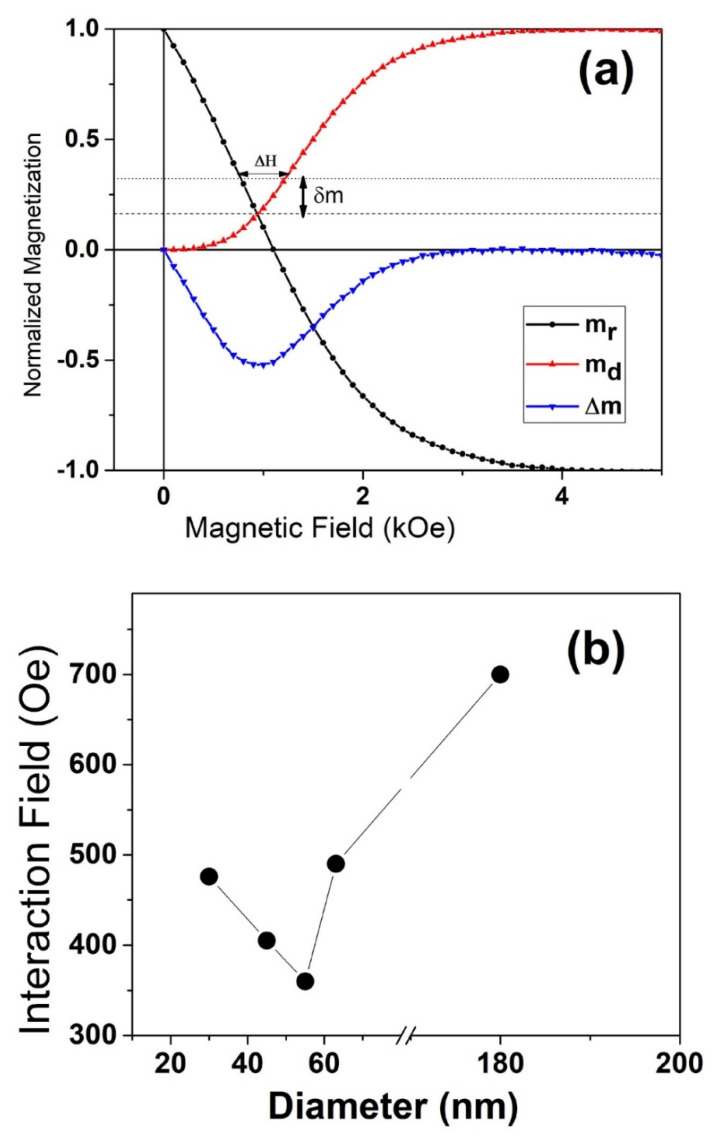
(**a**) m_r_ and m_d_ curves with the field-reflected DCD m_d_(*H*), and (**b**) estimated interaction field versus interwire spacing.

**Table 1 nanomaterials-11-03042-t001:** Anodization conditions and electrolyte used for AAO template fabrication. The anodization was carried out overnight at 0 °C. The time and temperature were kept same for all the samples.

Electrolyte	Concentration	Voltage	Diameter
C_2_H_2_O_4_	0.3 M	40 V	60 nm
C_2_H_2_O_4_	0.3 M	30 V	55 nm
C_2_H_2_O_4_	0.3 M	25 V	45 nm
H_2_SO_4_	0.3 M	20 V	30 nm

**Table 2 nanomaterials-11-03042-t002:** A summary of nanowires’ dimensions used in this study. Where δm = m_o_ − 1/3 and ΔH_1/3_ manifests interaction field among nanowires.

	Diameter D (nm)	Interwire Spacing S (nm)	Preferred Orientation	Ratio s/d	δm	Interaction Field $\alpha =\frac{3}{2}\;\Delta H_{1/3}$	P = 3.67 (d/s)^2^
HI	30	25	Parallel	0.74	0.13	714	5.3
H2	45	27	Perpendicular	0.61	0.13	608	10.2
H3	55	30	Perpendicular	0.55	0.11	548	12.3
H4	188	100	Perpendicular	0.53	0.0667	1050	14.7
H5	60	24	-	0.40	0.167	722	23

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
