# Peer review of "Tuning Easy Magnetization Direction and Magnetostatic Interactions in High Aspect Ratio Nanowires"

_nanomaterials, 2021, doi:10.3390/nano11113042_

Round 1

Reviewer 1 Report

The authors prepared arrays of hcp Co nanowires with various diameters and interwire spacings, and studied their microstructure and crystal structure using electron  microscopy and X-ray diffraction. The magnetic properties indicate an isotropic or 
anisotropic behavior depending on the interwire spacing to diameter ratio. The easy  magnetization direction can be tailored from the parallel to perpendicular direction of the nanowire axis growth.  It is found that  nanowires interact via dipolar coupling that acts as an additional uniaxial anisotropy favoring an easy magnetization axis perpendicular to the wire axis, and that easy axis of magnetization can be tuned parallel or perpendicular to the nanowire direction by changing their packing density. It is a very interesting  work and should be published in Nanomaterials. 

The reference sequences need to be adjusted in  both Lines 46 and 111 (5,6,1;6,16,17,1).  

Author Response

We are thankful to the reviewer for a thorough and detailed report. 

We thank the reviewer for pointing it out. As suggested by the reviewer, relevant references have been updated in the proper sequence. Moreover, the all the references are in the same style, as required by the Nanomaterials journal.

Reviewer 2 Report

The authors report the fabrication of Co nanowire arrays using anodized aluminum oxide (AAO) templates. Upon varying the spacing to diameter (s/d) ratio it was established that the easy magnetization axis can be tuned. In particular, in the case of small (s/d) ratios the easy magnetization axis becomes perpendicular to the nanowire axis. Decreasing further the (s/d) ratio an isotropic behavior is obtained. These findings are supported by hysteretic curves showing the switch in magnetization (parallel vs. perpendicular) as well as an isotropic behavior (s/d) from 0.74 down to 0.37. The magnetostatic interactions between nanowires are investigated using dc demagnetization (DCD) and isothermal remanence (IRM) techniques. The hpc structure of the Co nanowires is proved by XRD micrographs and the morphology of the nanowire layer is analyzed by SEM.

The topic presented here is of interest for nanotechnology, in particular for applications concerning next generation dense non-volatile storage. However certain aspects need to be clarified and some additional investigations are worth to be performed before the paper can be considered for publication:

1. The authors mention in Section 2 that "the nanowires length was kept the same". Then, in section 3, there is a statement regarding the change in the easy magnetization axis from parallel to perpendicular when the nanowire length increases. There is no detailed information regarding this behaviour with respect to nanowire length.   

2. The authors indicate that nanowires larger than 10 um exhibit a "magnetization frustration". The authors should detail this aspect.

3. It is mentioned that the shape of the nanowires impacts their magnetic behaviour. Particularly when the spacing is small, how do the geometrical effects impact the overall results?

4. The partial ordering in the nanowire array can also influence the magnetization behaviour. How would the authors characterize their samples (statistically) in terms of local ordering in the nanowire array?

5. For a more detailed interpretation of the magnetic interactions can the authors provide a FORC analysis?

Author Response

We are thankful to the reviewers for a thorough and detailed report. We are pleased to know the reviewers have found our article to be suitable for further consideration in Nanomaterials. We acknowledge the reviewers for their critical reading and giving several useful comments that indeed helped us to improve our manuscript further. We have addressed all the comments and criticism made by the reviewer and listed them below.

Our responses to suggestions/comments by the reviewers are addressed below. Reviewers’ comments are given in black where as our response is in blue font colour. Excepts from the revised manuscript are in italics.

Reviewer 3 Report

The authors have revised their manuscript taking into account some of the queries raised by all the referees, but there are still many sentences and mistakes in the re-submitted version of the work that need for being changed and amended in the text. Some examples are listed below:

Introductory section:

Page 2, line 46, references should be given in order to appearence (1, 5, 6).

Page 2, line 49, please use magnetocrystalline instead of magneto crystalline.

Page 2, line 60 use "of applications" instead of "of the applications"

Page 2, lines 67 to 70, DCD and IRM are defined in the first sentence, so, they do no need to be repeated again.

Page 2, line 75, what means (M/Ms)?, is the reduced or normalized magnetization, m?. Therefore, m = 1/3 is not previously well defined in the text.

Page 2, lines 79-81, the sentence still appears being confused, as previously highlighted, because the porosity term has not any sense for nanowires. This sentence should be properly changed by another more appropriated.

Section 2, Materials and methods

Page 3, lines 119-120: the following sentence has not any sense. Could the authors better explain what means pores diameter smaller than 30 V?

The authors could add a table with a list for all the electrolites and experimental conditions of the electrochemical anodizations (electrolytes, voltage, temperature, anodization times, etc) in order to clarify and guide to the reader for better understanding all the processes carried out during the sinthesis of Co nanowires with different diameters and lengths.

Could the authors explain why are they employing boric acid (H3BO3) during the electrodeposition of Co nanowires?

Section 3 Results and discussions: page 5, lines 180-187

The terms used in many sentences such as easy magnetization axis or easy magnetization direction should be employed in an uniform way thorough the entire text of the manuscript, instead of other confusing forms that still appear in some pharagraphs of the whole text of the manuscript, such as easy-axis magnetization, magnetization easy-axis, etc. The term "an easy ..." should be corrected.

Section 4 Conclusions: lines 275-277. The term nanowires porosity has no any sense in this context. It should be better replaced for another with more clear significance, as already explained.

Some references need to be revised in order to be fully completed, as e.g.: reference 15, where the number of pages is missing. Also, the references stile should be the same, concerning to the authors names, title of the work, etc.

In the figure 2 b) the number of scale of the bar scale given in the SEM image is missing. The inset of the SEM image for nanopores or nanowires at the bottom does not display any scale.

The vertical legend of the hysteresis loops displayed in figure 4 should be M/Ms or Normalized Magnetization instead of Magnetization. The horizontal legend should be changed into Magnetic Field (kOe) instead of Magnetic Field (KOe). Also the same for the figure 5 a).

Author Response

We are thankful to the reviewers for a thorough and detailed report. We are pleased to know the reviewers have found our article to be of suitable for further consideration in Nanomaterials. We acknowledge the reviewers for their critical reading and giving several useful comments that indeed helped us to improve our manuscript further. We have addressed all the comments and criticism made by the reviewer and listed them below.

Our responses to suggestions/comments by the reviewers are addressed below. Reviewers’ comments are given in black where as our response is in blue font color. Excerpts from revised manuscript are in italics.

Round 2

Reviewer 2 Report

The authors responded to most of my queries and updated the revised manuscript. Some of the issues raised are set to be addressed in a future work. However I find the improvements reasonable at the time and therefore I would recommend the paper for publication. 

Author Response

Response to reviewer’s comments

Manuscript : nanomaterials-1422911

Tuning easy magnetization direction and magnetostatic interactions in high aspect ratio nanowires

We are thankful to the reviewer for a thorough and detailed report. We are pleased to know the reviewer has found our article to be of suitable for publication in Nanomaterials. 

Sincerely,

Hafsa Khurshid, PhD

Reviewer 3 Report

The authors have revised their manuscript taking into account the queries raised by the referees, therfore, this work can be considered for its publication in journal in the present form.

Author Response

Response to reviewer’s comments

Manuscript : nanomaterials-1422911

Tuning easy magnetization direction and magnetostatic interactions in high aspect ratio nanowires

We are thankful to the reviewer for a thorough and detailed report. We are pleased to know the reviewer has found our article to be of suitable for publication in Nanomaterials. We acknowledge the reviewers for their critical reading and giving several useful comments that indeed helped us to improve our manuscript further. We have addressed all the comments and criticism made by the reviewer.

Sincerely,

Hafsa Khurshid, PhD
